



# Model-free Estimation of Available Power using Deep Learning

Tuhfe Göçmen[1], Jaime Liew[1], Albert Meseguer Urban[1], and Alan Wai Hou Lio[1]

[1]Department of Wind Energy, Technical University of Denmark, Frederiksborgvej 399, DK-4000 Roskilde

**Correspondence:** Tuhfe Göçmen (tuhf@dtu.dk)

**Abstract.** In order to assess the level of power reserves during down-regulation, the available power of a wind turbine needs to be estimated. The current practice in available power estimation is heavily dependent on the pre-defined performance parameters of the turbine and the curtailment strategy followed. This paper proposes a model-free approach for a single input, dynamic estimation of the available power using recurrent neural networks. The unsteady patterns in the turbulent flow are represented via Long Short-Term Memory (LSTM) neurons which are trained during a period of normal operation.

The model-free approach requires only 1-Hz wind speed measurements as the input and generates 1-Hz available power estimation as the output. The neural network is trained, tested and validated using the DTU 10MW reference wind turbine HAWC2 model under realistic atmospheric conditions. The adaptability of the network to changing inflow conditions is ensured via transfer learning, where the last LSTM layer is updated using new measurements. It is seen that the sensitivity of the networks to changing wind speed is much higher than that of turbulence, and the updates are to be implemented solely based on the altering inflow velocity. The validation of the trained LSTM networks on time series with 7, 9 and 11 m/s mean wind speeds demonstrates high accuracy (less than 1% bias) and capability of transfer-learning. Including highly turbulent inflow cases, the networks have shown to easily comply with the most recent grid codes, which require the quality of the available power estimations to be evaluated with high accuracy (less than 3.3% standard deviation of the error) at 1-min intervals.

## 1 Introduction

As the share of wind energy increases in power systems around the world, new challenges regarding the control and operations of a power system are encountered. In order to maintain power system stability, transmission system operators (TSOs) are developing new grid codes requiring contributions not only from conventional generators, but also from wind power plants (see *e.g.* Ministere des Affaires Economiques (2019); Energinet.dk (2017); 50Hertz, Amprion, Tennet, TransnetBW (2016); Elia, Belgium (2015); EirGrid (2015); National Grid (2014)). In this context, the paper focuses on the active power contribution of wind turbines to provide frequency support via down-regulation (also referred to as curtailment, de-rating or de-loading).

The curtailment of wind turbines can be implemented either as balance control, where the turbine power output is reduced to a constant value, or as delta control, where the output is reduced by a certain percentage of the available power (Attya et al., 2018; Fleming et al., 2016; Hansen et al., 2006). Additionally, the power output reduction for both strategies can be achieved by adjusting the rotor blade pitch angles or operating at a sub-optimal rotor speed compared to the maximum energy capture value (Wilches-Bernal et al., 2016). Although modern turbines are capable of implementing both balance and delta control,



due to the uncertainties in the estimated available power (Göçmen et al., 2019; Göçmen and Giebel, 2018; Göçmen et al., 2016; Pinson et al., 2007; Pinson, 2006), balance control is the preferred industrial application as its set-point is independent of the available power (Kristoffersen, 2005). The amount of power reserves however, which is defined as the difference between the
available and the produced power under curtailed operation, does depend on the available power in the wind for both delta and balance control. This is particularly critical for wind farm developers to get compensation under mandatory down-regulation, as well as (existing and expected) flexible balancing market structures, where the reserve power is traded at different time scales depending on the regional balancing market schemes (Chinmoy et al., 2019).

Generally, trading in the electricity markets is performed in advance with a given forecast horizon. Depending on the bidding
structure, the available power production of an asset is to be predicted sometimes as short as 5-min ahead (*e.g.* Rana and Koprinska (2016)). The forecasting tools can be based on physical or statistical modelling, as well as the combination of both. Many perform post-processing via model output statistics to reduce the remaining error. Some approaches focus on the best possible estimate of the local wind speed while some directly extract the wind power generation potential. Statistical models use explanatory variables and historical/online information (measurements, log-data, etc.), generally implementing recursive
techniques, such as recursive least squares or artificial neural networks (or deep learning) (Sideratos and Hatziargyriou, 2007). In fact, forecasting is the field with the most deep learning (and broadly artificial intelligence) applications in wind energy (Ata, 2015). However, while forecasting the available power the operational status and potential effects of control scenarios are often overlooked, especially for higher (than *e.g.* 5-min) frequencies at a single turbine level.

For the operational considerations and higher frequency system stability issues, the time scales considered in the market-
based forecasting are already long-term ahead. In order for the balancing responsible parties to get compensated during manda-tory down-regulation by the TSOs, wind power plants are expected to provide information regarding their power production in much shorter time scales. As stated in the recent grid requirements in Germany (50Hertz, Amprion, Tennet, TransnetBW, 2016), the available power is to be calculated for 60-seconds intervals for down-regulated wind farms. Additionally, the 1-minute standard deviation of the percentage error of the available power is required to be less than $\pm$ 3.3% (after the pilot
phase). The enforced regulations are difficult to comply and are subject to penalty if not met.

The current practice in available power estimation is to assess the incoming wind speed to derive the possible power output of the turbine via optimum performance curve. One of the most common approaches to approximate the (effective) wind speed is by solving the static wind power equation, which is widely adopted in the wind turbine industry as well as the wind research communities (*e.g.* (van der Hooft and van Engelen, 2004; Göçmen et al., 2014)). More details on the approach is provided
in Section 2. (Ma et al., 1995) demonstrates that directly mapping the static relation does not give satisfactory performance and concludes that the inclusion of dynamic models can significantly improve the wind speed estimate. Thus, an increasing number of studies (*e.g.* (Østergaard et al., 2007; Meng et al., 2016)) began to utilize observer theory, in particular, Kalman filtering. For example, in (Østergaard et al., 2007), the aerodynamic torque is considered as a system disturbance state, and it is estimated by the use of an observer-based system on a simple drive-train model with pre-defined dynamics for the aerodynamic
torque. Subsequently, the calculation of the wind speed is done by inversion of the static mapping between the aerodynamic torque and wind speed. One of the drawbacks of searching through the static relation, to find the wind speed estimate, is its





computationally cost, where a Newton-Raphson method is often employed to find the corresponding wind speed given the turbine measurement on a discrete power coefficient $C_p$ surface. On the other hand, some methods do not require the use of iterative gradient methods, for example, by considering the wind speed directly as a state to the system and such a wind state

can be estimated via an observer/Kalman filter. In (Selvam, 2007), the wind dynamics are modelled as a random walk and augmented with a linear turbine model including a simple drive-train and tower dynamic model. A linear Kalman filter is then employed to estimate the wind speed for feed-forward control purpose. Similar techniques also have been utilised in (Stol and Balas, 2003; Simley and Pao, 2016). A study by (Knudsen et al., 2011) employed a non-linear turbine model including a simple drive-train, tower and wind speed dynamics where the effective wind speed is estimated by an extended Kalman filter . Similar

methods are also reported in (Henriksen et al., 2012), where dynamic inflow model is included. Besides the Kalman filter-based approaches, some studies (Ortega et al., 2011, 2013) used a more advanced state estimation technique of immersion and invariance to construct a wind speed estimation with proof of global convergence under certain assumptions. For more detail and further information on wind speed estimation, see (Soltani et al., 2013) and references therein.

The state-of-the-art available power estimation is heavily dependent on the considered turbine models, as well as the op-
eration strategy for curtailment. More specifically, the majority of the methods rely on the pre-calculated power coefficient, $C_p$, or the certified nominal power curve to convert (rotor-effective) wind speed to (available) power. However, the varying wind speed and turbulence levels activate different dynamics within the turbine structure and cause different control responses (Murcia et al., 2018). In addition, temporally and spatially local characteristics of the flow (*e.g.* humidity, temperature, etc.) and the condition of the turbine (*e.g.* blade erosion, dust, component wear or failure, etc.) highly affect the $C_p$ and the power curve

behaviour. Therefore, these generally deterministic approaches fail to represent the detailed dynamics required to produce high frequency available power signal accurately (Jin and Tian, 2010), and they are an important source of uncertainty (Lange, 2005). In order to tackle the inadequacy of the turbine models to fully represent the dynamic power output of a turbine under turbulent inflow, a model-free approach to transfer the wind speed to power is a strong alternative.

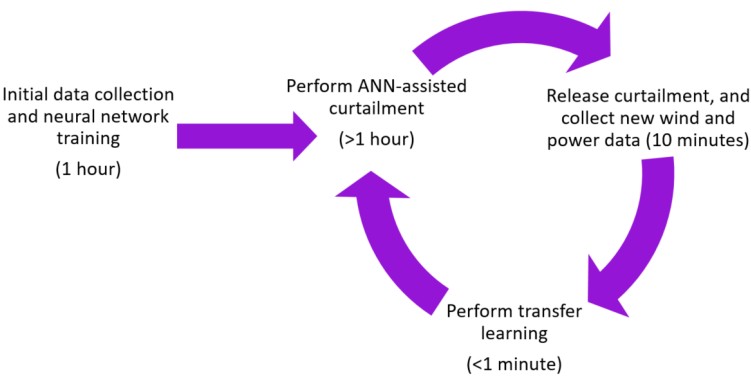

**Figure 1.** Flow diagram of the model free approach to estimating available power.



Therefore in this study, the aim is to bring the wind turbine generator (WTG) control considerations and modern forecasting
methodologies for a model-free, single input estimation of available power. It enables a robust real-time implementation of
dynamic delta control, as well as the provision of the reserves to the system level within the frame of (strictest) European grid
regulations. In the model-free estimation of available power, the unsteady patterns in the turbulent flow is represented via Long
Short-Term Memory (LSTM) Hochreiter and Schmidhuber (1997) neurons, which is a special building unit for Recurrent
Neural Networks (RNNs). The proposed method for integrating an LSTM network in a curtailment strategy is outlined in
Figure 1. During a period of normal operation of a WTG, wind speed and power output time series data is collected for at least
an hour to establish a training data set. Next, the network is trained on the collected data which, depending on the available
processing power, can be performed within seconds. The wind turbine operator can then announce its participation in the
reserve market online or ahead of time with the intention of performing delta or balance control. Down-regulation is then
performed using the LSTM predictor to provide the set point of the available power. The model-free estimation approach can
be rapidly retrained with newly collected data using transfer learning, where the last LSTM layer of the network is updated
using the new information.

The synthetic time series used in this study is generated using the DTU 10MW reference wind turbine model with the
aeroelastic code HAWC2 (Bak et al., 2012) under realistic atmospheric conditions, and the simulation results are publicly
available[1]. First, the sensitivity of the state-of-the-art available power predictions to the curtailment operation strategy is briefly
discussed and quantified in Section 2. To address the issue, a detailed analysis of LSTM neural networks and the potentials of
transfer learning to adapt changing inflow conditions is presented throughout Section 3. This research focus is highly important
for the individual turbine control and its role in the power system stability, as well as the business case of wind energy in the
existing and upcoming market scenarios.

## 2   Wind speed-to-Power via Turbine Model

As stated earlier, current methods for estimating available power typically make use of pre-defined nominal operation power
curves or power coefficient calculations. Here in this section, we discuss the assessment of wind speed and the sensitivity of
the model-dependent approaches to the implemented curtailment strategy.

Point measurements of the wind speed using, for example, cup or sonic anemometers, are often unreliable at estimating the
potential power production of a wind turbine as the spatial variations in the wind field are not captured. For example, a naive
approach at estimating available power is:

$$P_{avail}(U) = \frac{1}{2}\rho\pi R^2 C_p U^3 \tag{1}$$

where the air density, $\rho$, rotor radius, $R$, and power coefficient, $C_p$ are assumed to be constant. Equation (1) presents a number
of weaknesses, namely the inability to capture the dynamic response of the wind turbine to changing wind speeds, or the
spatial variations in the wind field. For this reason, the rotor effective wind speed, which is defined as the spatial average wind

---

[1]The generated time series can be accessed here: https://gitlab.windenergy.dtu.dk/tuhf/deep-learning-for-available-power-estimation/tree/master/data





speed over the rotor plane, is preferred in terms of power estimation. Although there are numerous methods for estimating rotor effective wind speed, the majority of methods use operating data of the wind turbine to create the estimate (Jena and Rajendran, 2015). A simple strategy is to estimate the wind speed for a given power output using a polynomial fit (Thiringer and Petersson, 2005). This method can be extended by including the rotor speed and blade pitch angle in conjunction with a $C_p$ look up table to infer the wind speed as shown in Bhowmik et al. (1998). For derated operation, these methods are problematic

as the dependency between the wind speed and a turbines operating points vary based on the desired level of down-regulation. The use of a predefined $C_p$ curve to estimate available power therefore becomes unreliable. However, state-space approaches where the convergence of the wind estimation error is analysed systematically can potentially respond to that problem.

  There are several benefits of formulating a wind speed estimation as a system state estimation problem compared to methods that use static relation mapping the power or aerodynamic torque to wind speed. For example, a substantial body of mature and sophisticated state estimation theory can immediately be brought to bear upon the design of the wind estimator. Moreover,

in an observer design where the wind speed is considered as a system state, the use of slow gradient methods for solving the static relations can be avoided, resulting in better computational speed and a smooth wind speed estimate.

  Typically, to formulate a state estimation problem, a simplified model of the non-linear dynamics is required that needs to capture the key dynamics of the turbine. For brevity, a widely used non-linear turbine system model is employed, including the

dynamics of rotor drive-train, tower and wind speed (See Knudsen et al. (2011); Lio et al. (2019)):

$$x_{k+1} = f(x_k, u_k) + w_{\mathrm{n},k}, \tag{2a}$$
$$y_k = h(x_k, u_k) + v_{\mathrm{n},k}, \tag{2b}$$

where $x_k = [\omega_k, \dot{x}_{\mathrm{fa},k}, x_{\mathrm{fa},k}, v_k]^T \in \mathbb{R}^{n_x}$ is the system state vector containing the rotor speed, fore-aft velocity and displacement of the tower-top and ambient wind speed, whilst the system input $u_k = [\tau_{g,k}, \theta_k]^T \in \mathbb{R}^{n_u}$ contains the generator torque and

pitch angle and $y_k = [\omega_k, \dot{x}_{\mathrm{fa},k}, x_{\mathrm{fa},k}]^T \in \mathbb{R}^{n_y}$ denotes the system output. The state transition and output functions are denoted as $f : \mathbb{R}^{n_x} \times \mathbb{R}^{n_u} \to \mathbb{R}^{n_x}, h : \mathbb{R}^{n_x} \times \mathbb{R}^{n_u} \to \mathbb{R}^{n_y}$. The Gaussian process noise $w_{\mathrm{n},k} \in \mathbb{R}^{n_x}$ represents the modeling errors whilst the Gaussian measurement noise $v_{\mathrm{n},k} \in \mathbb{R}^{n_y}$ represents the sensor noise and modelling error of the sensor dynamics.

  Since the turbine model is a nonlinear model (2a), an extended Kalman filter (EKF) is employed to compute estimates of the wind turbine state. A Kalman filter is a computationally efficient and recursive algorithm that provides the optimal state

estimates $\hat{x}_k \in \mathbb{R}^{n_x}$ by minimising the mean square state error or the state error covariance matrix $P_k := E[(x_k - \hat{x}_k)(x_k - \hat{x}_k)^T]$. Kalman filtering approaches have been effectively employed in many examples of wind energy (e.g. Ritter et al. (2018); Lio (2018); Annoni et al. (2018)). Typically, in EKF, the estimate of the state $\hat{x}_k$ is computed in two-step processes: prediction and measurement update. The superscripts $x_k^+, x_k^-$ are denoted as the variable $x$ at sample time $k$ after the measurement update and before the measurement update, respectively.





Prediction :

$$\hat{x}_k^- = f(\hat{x}_{k-1}^+, u_k), \ \ P_k^- = F_k P_{k-1}^+ F_k^T + Q_k, \ \ F_k := \frac{\partial f(\hat{x}_{k-1}^+)}{\partial x}, \tag{3a}$$

Measurement update :

$$\hat{y} = h(\hat{x}_k^-, u_k), \ \ \hat{x}_k^+ = \hat{x}_k^- + L_k(y_k - \hat{y}_k), \ \ P_k^+ = (I - L_k H_k)P_k^-, \ \ H_k := \frac{\partial h(x_k^-)}{\partial x}, \tag{3b}$$

where $L_k \in \mathbb{R}^{n_x \times n_y}$ is the filter gain and it is computed as follows:

$$L_k = P_k^- H_k^T (H_k P_k^- H_k^T + R_k)^{-1}, \tag{3c}$$

where $Q_k \in \mathbb{R}^{n_x \times n_x}$, $R_k \in \mathbb{R}^{n_u \times n_u}$ denote the co-variance matrices of the process and measurement noises, respectively, that can be computed as $Q_k = E[w_{n,k} w_{n,k}^T]$, $R_k = E[v_{n,k} v_{n,k}^T]^T$. The process co-variance $Q_k$ is chosen by approximating the variance of the modelling error and the typical wind speed. In this work, there is no measurement noise, thus, the measurement co-variance $R_k$ is chosen as a small value.

One of the weaknesses of the EKF filtering approach is being a model-based method, that requires relatively an accurate model. Besides, the choice of the model, operating conditions and sensor locations also strongly affect the EKF-based estimator performance (Lio et al., 2019). Some studies (*e.g.* Lio et al. (2018)) showed that down-regulations can be achieved by either modifying the generator torque solely or the combinations of rotor speed and torque. The constant and maximum rotation (Const-$\Omega$ and Max-$\Omega$) strategies perform down-regulation by setting the rotor speed to a pre-determined or maximum
value, respectively, whilst the min-$C_t$ methods operate the turbine at mimimum thrust coefficient in down-regulations. The performances of the EKF based upon these operations are shown in  2. The simulations are based on DTU 10MW reference wind turbine HAWC2 model (Bak et al., 2012) under 9 m/s mean wind speed and 10% mean turbulence intensity (TI) over 700 seconds. The turbines are commanded to operate at 40% and 80% of the rated power. One clear message from Figure 2 is that the performance of the EKF-based wind estimator is heavily subjected to the turbine operating conditions, for example,
the performances were similar for strategies operating at 80% but the Max-$\Omega$ performed the worst at 40% down-regulation. Similarly, Min-$C_t$ shows the best agreement with the available power for 40% downregulation whereas it performs the worst for 80% curtailment. Therefore, Figure 2 indicates no clear trend and high sensitivity of model-based methods to the control scenario.

It should be noted that, the sensitivity observed in the synthetic time series in Figure 2 is expected to grow under the field
conditions. This is due to the fact that the manufacturer-calibrated power coefficients cannot account for variability influenced by local conditions (Bandi and Apt, 2016). Additionally, the resulting uncertainty of the $C_p$ dependent approaches is likely to be amplified also due to the lack of detailed information regarding the pre-defined $C_p$ and implemented operation strategy for curtailment caused by the limited access to the controller in practice. To avoid the dependency on operating point estimations of available power, the use of wind speed measurements is revisited with the state-of-the-art deep learning architecture in the
next section.





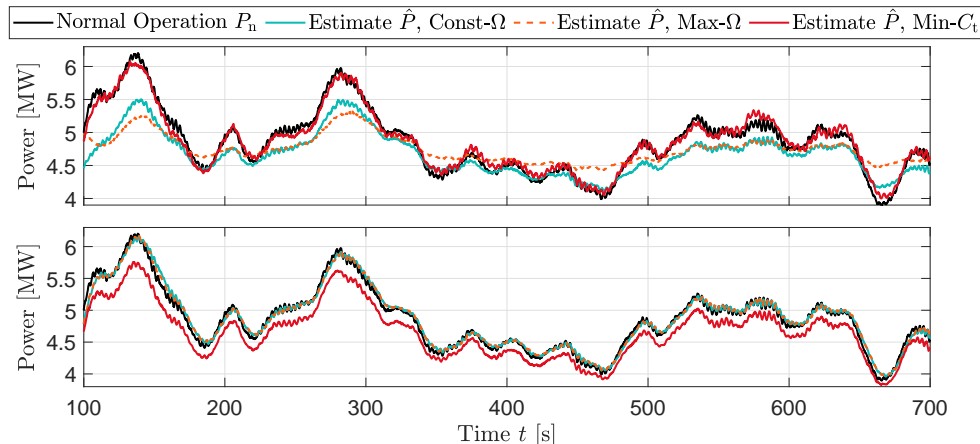

**Figure 2.** Time series of normal power $P_\mathrm{n}$ and available power estimation $\hat{P}$ based upon various down-regulation strategies. The top and bottom plots indicate the estimation based on measurements of turbines operating at 40% and 80% of the rated power, respectively.

## 3 Neural Networks for Available Power set-point

For a more robust operation and delta control, the bias and the uncertainties which is partly originated from the the natural variability of the flow and turbulence, and partly due to the uncertainty associated with the turbine models *i.e.* $C_P$ surfaces, should be reduced. The former is investigated through a state-space update via Kalman filters in Section 2. Here, we implement a fully data driven approach, which is purely based on the atmospheric inputs to eliminate the dependency of the estimated available power production to the $C_P$ surfaces and/or the control strategy.

Although the deep learning techniques have been applied to numerous engineering fields, their application in wind farm flow modelling has been rather limited. For the turbine level power estimation, recently neural networks have been implemented to approximate the power curve mainly based on field data (for a detailed review, see *e.g.* Lydia et al. (2014)). Pelletier et al. (2016) applied feed forward neural networks (FFNNs) in a steady-state manner with 6 atmospheric inputs including shear and yaw error of the investigated turbine. Ouyang et al. (2017) approached to the problem by sectioning the regions of the power curve and developed a support vector machine algorithm for each partition, capable of capturing the dynamic response of the turbine. Manobel et al. (2018) on the other hand, underlines the importance of data filtering and normal behaviour recognition for such problems and also indicates that the architecture of the neural network needs to be re-optimised for each turbine within a wind farm, to increase accuracy.

Here in this study, we use the open-source machine learning repository, TensorFlow (Abadi et al., 2016) to implement the LSTM algorithm (Hochreiter and Schmidhuber, 1997). LSTM architecture is a special type of RNN, which is shown to perform faster and better for highly fluctuating time series than many other RNN architectures. An LSTM neuron is illustrated in Figure 3, where there is no direct connection between the input $i_t$ and the output $o_t$ gates. All the information flows through the cell state $c_t$, which is the actual memory of the LSTM neuron and it is regulated by the forget gate $f_t$ to avoid indefinite





growth and eventual network break down (Gers et al., 2000). Through the calibrated weights, $f_t$ decides how much of the previous cell state(s) is preserved. LSTM algorithms are heavily used in a variety of sequential/temporal predictive modelling, from language processing (*e.g.* Gers and Schmidhuber (2001)) to short-term forecasting (*e.g.* Zhang et al. (2019)). However, they require large amounts of data and computational resources to reach their full potential and achieve a generic solution without over-fitting. Therefore, although RNNs (and LSTMs in particular) have additional capabilities of modelling longer-term temporal properties, they remain highly challenging to train especially with limited training data. In recent years, the transfer learning (or knowledge transfer) approach that addresses such problems (Pan and Yang, 2010) has been increasingly popular. The basic idea of the transfer learning is that a well-trained model and its hyper-parameters that involve rich knowledge of the target task can be used to guide the training of other models.

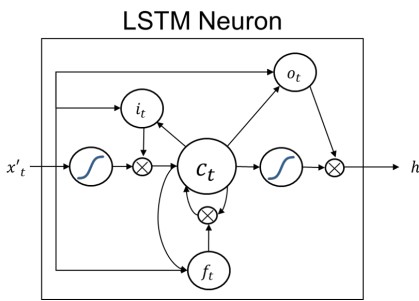

**Figure 3.** LSTM Neuron with cell state $c_t$, as well as input $i_t$, output $o_t$ and forget $f_t$ gates. $x'_t$ and $h_t$ indicate the inputs and outputs of the neuron, respectively.

Throughout the rest of this section, we will firstly present the details of the architecture and the hyper-parameter tuning of an LSTM model fully trained on 3-hours of HAWC2 simulations. We will then challenge our model to perform on another case with a different inflow condition than the original training domain. In pursuit of better performance on a different flow case, we will present the results from blind training as well as the transfer learning. We will then discuss their behaviour both in terms of the resulting error distributions and fitted parameters in between the layers. Finally, we will extend the application of the transfer learning to other flow cases, to demonstrate the flexibility and automation of the approach.

### 3.1 Data Pre-Processing and Training Strategy

The investigated case studies for available power estimation is generated and implemented using HAWC2 simulations with DTU 10MW reference wind turbine. For the training of the LSTM models, the high frequency (100Hz) wind speed signals from HAWC2 is down-sampled to 1-Hz, which is equivalent to what can typically be extracted from the turbine SCADA systems (Göçmen and Giebel, 2018). The second input to the model is a moving (or rolling) standard deviation of the 1-Hz wind speed, with a 10-min rolling window as an indication of turbulence intensity (TI) in the inflow. In contrast to the regular definition, this approximation of TI assures the same number of samples for both of the inputs.





The two inputs of wind speed and its moving standard deviation are first normalized between (0, 1) and then fed to the LSTM network to predict the power output during normal operation. As an LSTM neuron expects a 3-dimensional input shape

220 in the order of samples, lag and features, the input data is shaped accordingly. For the defined architecture, the number of (input) features have been listed as 2 (wind speed and moving TI) and the lag, which is the hindsight horizon to base the real-time predictions on, is another hyper-parameter to be tuned. For all the trained networks the lag of 4s, 9s, 29s, 59s and 89s are investigated. Note that since the model is trained to map the atmospheric inputs to the actual production data under normal operation, the power predictions are ensured to follow the normal operation trend that is required for the available

225 power estimation and not affected by the curtailment strategy.

For the preliminary evaluation of the training and hyper-parameter tuning of the model, a split test dataset is generated. The final validation of the model is based on an independent time series with a similar mean wind speed and turbulence intensity, but covers a shorter period. Since the target application of the model is to estimate real-time available power for more certain delta control (or reserve provision), the main criteria of evaluation is 1-Hz error distribution.

230 ### 3.1.1 Training of the First LSTM Model : Low Wind Speed, High Turbulence Intensity

The case study to train the first LSTM model consist of 3-hours period, where hub-height wind speed and corresponding moving TI are used to estimate the power output of DTU 10MW turbine under nominal operation. The input time series are presented in Figure 4.

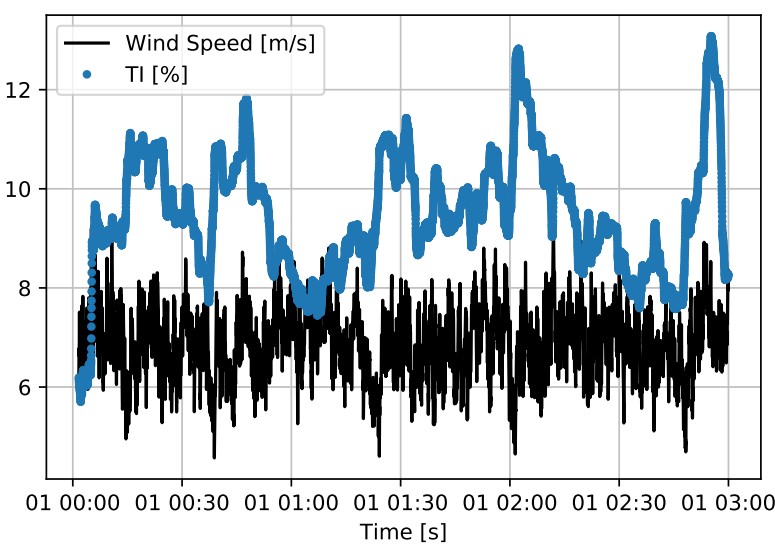

**Figure 4.** First LSTM model training input time series generated by HAWC2, down-sampled to 1-Hz. Mean Wind Speed = 7 m/s, Mean TI = 10%.





In order to have an adequate quantity of training samples while assuring a representative test dataset for hyper-parameter tuning, the 3-hour period of training and testing signals are split as $80\% - 20\%$, respectively. Since the target available power output is 1-Hz, the final model needs to be able to handle low frequency dynamics in the inflow and successfully map it to the produced power in normal operation, by taking the inertia into account. Given the complexity the model is required to manage, the final network has 3 hidden layers with 100, 100 and 40 LSTM neurons, as indicated in Figure 5. The hyperbolic tangent function is used as the activation function in between the layers. With the listed input structure, the final architecture corresponds to approximately 7 times more data than the trainable parameters, slightly less than the general rule of thumb to avoid overfitting. However, the training history and the model performance on the test dataset does not indicate a clear overfit, increasing confidence to the training (see Appendix A, Figure A1 for training history).

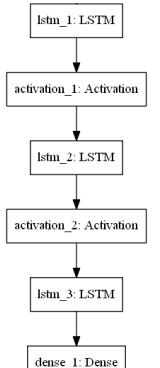

```
Layer (type)                    Output Shape              Param #
=================================================================
lstm_1 (LSTM)                   (None, 30, 100)           41200
_________________________________________________________________
activation_1 (Activation)       (None, 30, 100)           0
_________________________________________________________________
lstm_2 (LSTM)                   (None, 30, 100)           80400
_________________________________________________________________
activation_2 (Activation)       (None, 30, 100)           0
_________________________________________________________________
lstm_3 (LSTM)                   (None, 40)                22560
_________________________________________________________________
dense_1 (Dense)                 (None, 1)                 41
=================================================================
Total params: 144,201
Trainable params: 144,201
Non-trainable params: 0
```

**Figure 5.** Architecture of the First LSTM model for lag = 29s. *Hidden Layers:* `lstm_1`, `lstm_2`, `lstm_3`. *Output Layer:* `dense_1`. $\tanh$ is used as the activation function in both of the activation layers. The output of the hidden layers is in the shape of (time steps, lag, number of neurons in that layer.)

The first LSTM model is validated on separate 10-min dataset and compared with the *true* Available Power (actual production of the DTU 10MW turbine under normal operation) as well as the predecessor method of pre-defined $C_P$ look-up tables of the same turbine. The 1-Hz time series and the corresponding 1-second percentage error distribution of the direct $C_P$ look-up table approach and the LSTM model results with lag = 29s are presented in Figure 6. The sensitivity of the mean, $\mu_{\text{LSTM}}$, and the standard deviation, $\sigma_{\text{LSTM}}$ to the hindsight horizon up to 89s is listed in Table 1.

The hindsight horizon, or lag, is the number of previous time steps that have been taken into account to predict the power output in the current time step. Note that longer lag would increase the initialization period for the curtailment implementation and could be a limiting factor if the architecture is to be further adapted for online learning/training. Due to highest overall performance, the results from LSTM model with lag = 29s will be discussed from now on.

Figure 6 shows that the LSTM model significantly improves the agreement between the actual and the predicted available power production compared to the direct $C_P$ curve interpolation. Since both the trained LSTM model and the $C_P$ curve interpolation approach uses the same input (hub-height wind speed), it can be said that the described deep learning architecture is much more capable of reproducing the dynamic power curve of the turbine than the steady-state $C_P$ surface, even with





limited information. For the investigated 10-min period, the bias in the second-wise LSTM available power predictions is essentially zero, as opposed to nearly 7% observed with the direct $C_P$ interpolation approach. The error distribution is close to Gaussian distribution, which provides simpler post-processing options especially when the available power prediction is to be delivered at larger time scales (*e.g.* Gaussian averaging of the LSTM model results for reduced uncertainties within the

minute scale as requested by the German TSOs (50Hertz, Amprion, Tennet, TransnetBW, 2016)). The standard deviation of the second-wise error distribution, which is regarded as indication of uncertainty in the model results for this study, is further reduced with the Gaussian convolution filter. The mean error of the 1-Hz available power prediction remains less than 1%, where the percentage error is defined as $\frac{y(t_i)-\hat{y}(t_i)}{y(t_i)}100$ with $y(t_i)$ being the power produced by DTU 10MW under normal operation, *i.e.* available power, and $\hat{y}(t_i)$ is the LSTM model prediction at every time step $t_i$.

| Lag | $\mu_{LSTM}$ [%] | $\sigma_{LSTM}$ [%] |
|-----|------------------|---------------------|
| 4s | 2.37 | 13.4 |
| 9s | 1.68 | 13.37 |
| 29s | -0.34 | 11.66 |
| 59s | -1.02 | 12.42 |
| 89s | 0.18 | 12.68 |

**Table 1.** Sensitivity of the First LSTM model to the hindsight horizon.





(a) 1-Hz time series of available power and wind speed for the First validation case.

(b) Available power estimation error via direct $C_P$ curve interpolation of wind speed.

(c) Error distribution of the First LSTM model, 'raw' results.

(d) Error distribution of the First LSTM model, with Gaussian convolution.

**Figure 6.** Second-wise comparison of the Available Power of the 10MW turbine for mean wind speed = 7 m/s and mean TI = 10% flow case represented in Figure 4. $\mu$ and $\sigma$ are the mean and the standard deviation of the 1-Hz percentage error distributions for direct $C_P$ curve interpolation approach and the First LSTM model with lag=29s.

### 3.1.2 Training of the Second LSTM Model : High Wind Speed, High Turbulence Intensity

One of the most crucial challenges of purely data driven models is the fact that they are not valid for the input variables outside the training domain, also referred as generalization problem. As seen in Figure 4 the First LSTM model is trained for mean wind speed 7 m/s, where the turbulent fluctuations occasionally reach above 8 m/s. However, for higher wind speeds *e.g.* around 9 m/s, the First LSTM model is expected to perform poorly as it has not been taught to map the relationship between wind speed, TI and Power for that inflow.




In order to reduce the effort in hyper-parameter tuning and test the universality of the network architecture for a similar problem, the same model structure as in Figure 5 is implemented with the inflow time series presented in Figure 7. The final model is referred as the Second LSTM model throughout this study.

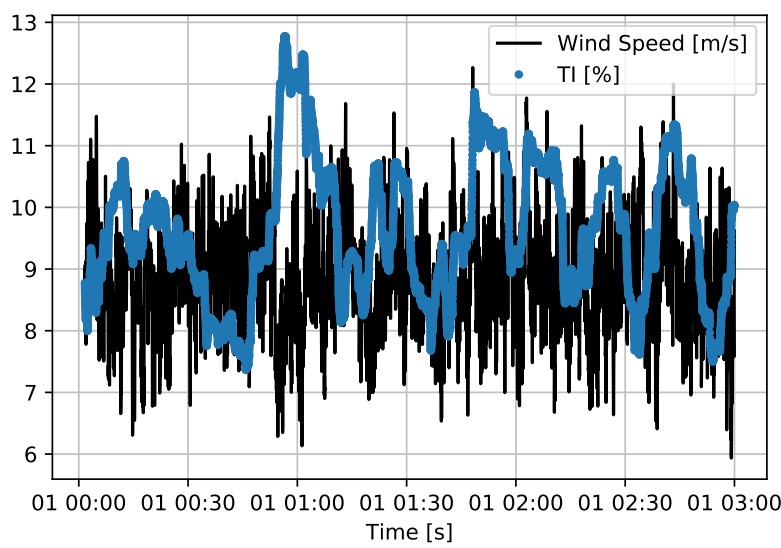

**Figure 7.** Second LSTM model training input time series generated by HAWC2, down-sampled to 1-Hz. Mean Wind Speed = 9 m/s, Mean TI = 10%.

The performance of the Second model is evaluated based on an independent 10min series with similar mean wind speed
and TI and compared with the direct $C_P$ interpolation approach. Similar to the First LSTM model, the validation results are presented for lag=29s in Figure 8.

| Lag | $\mu_{LSTM}$ [%] | $\sigma_{LSTM}$ [%] |
|-----|-----|-----|
| 4s  | 3.18 | 13.32 |
| 9s  | 2.71 | 12.27 |
| 29s | 3.59 | 12.13 |
| 59s | 3.0  | 12.24 |
| 89s | 2.85 | 12.84 |

**Table 2.** Sensitivity of the Second LSTM model to the hindsight horizon, 'raw' results





**(a)** 1-Hz time series of available power and wind speed for the Second validation case.

**(b)** Available power estimation error via direct $C_P$ curve interpolation of wind speed.

**(c)** Error distribution of the Second LSTM model, 'raw' results.

**(d)** Error distribution of the Second LSTM model, with Gaussian convolution.

**Figure 8.** 1-sec comparison of the Available Power of the 10MW turbine for mean wind speed = 9 m/s and mean TI = 10% flow case represented in Figure 7. $\mu$ and $\sigma$ are the mean and the standard deviation of the 1-Hz percentage error distributions for direct $C_P$ curve interpolation approach and the Second LSTM model with lag=29s.

Despite the significant performance improvement achieved for 7 m/s case with the First LSTM observed in Figure 6, the Second LSTM model developed using the same procedure for 9 m/s inflow has a considerable bias of more than 3% as seen in the 1-Hz percentage error distribution in Figure 8. The mean and the standard deviation of the model error seems to be
hardly affected by the changing hindsight horizon listed in Table 2. This clearly implies that the architecture and the hyperparameters optimized for the lower wind speed are not necessarily the best configuration for slightly higher wind speed cases. That trend makes it challenging to develop a generic network architecture that would successfully reproduce the high frequency available power for all the possible input realizations. It indicates the need to specifically tune the hyper-parameters for each separate flow case. It is a cumbersome process with very little room for automation. Here in this study, the focus is to make
the best out of the available dataset, as indicated earlier, as the generation (or collection) of a comprehensive database is a





very demanding task for high frequency problems. Additionally, the observed reduction in performance of the same hyper-parameter space for a different flow case indicates the risk of the approach where a singular 'generic' model is fit to estimate the high frequency available power for a variety of inflow cases. In other words, a single model to cover the entire domain might introduce compromises in the model performance at certain inflow cases, where the dynamic accuracy is of utmost importance,

as framed by the grid codes.

### 3.1.3 Transfer Learning from the First Model : High Wind Speed, High Turbulence Intensity

Having trained a well-performing model for first inflow case with 7 m/s mean wind speed and 10% mean TI, the question arises: Can some of the characteristics of the First Model be conveyed to a different flow case to achieve similarly good results? Transfer learning can provide a valuable platform for such model extensions, as it is used to improve a learner from

one domain by transferring information from a related domain (Weiss et al., 2016). This enables a systematic model update when new data is available from outside the training domain. Accordingly, part of the First Model with 7 m/s mean wind speed would be transferred to update some of the parameters for higher wind speed. The procedure could be repeated for all the changing wind speed and TI cases, both in HAWC2 platform and field applications.

To assess the transferability of the parameters, the trends of the weights trained for the First (Model_1) and the Second

(Model_2) LSTM model is compared in Figure 9. The actual probability seen in the most recent histograms (the lightest shade in the series of distributions) are different for all three LSTM layers, with larger tails on Model_1 distributions. However, the range of values for the output weights of the first layer lstm_1 and the second layer lstm_2 are very similar, with interquartile range -0.05 < IQR < 0.05 for both. On the other hand, the third and shallower layer lstm_3 seem to optimize for significantly different weights for different inflow velocities. Therefore, it is concluded that the first 2 LSTM layers are transferable from the

First Model, where the last LSTM layer as well as the output layer need to be re-tuned for changing inflow case(s). The resulting architecture is presented in Figure 10 where the number of trainable parameters are significantly reduced. Accordingly, the transferred architecture is a much lighter network that ensures fast training, while enclosing a profound amount of information from previous learning(s). Less number of parameters also enables a robust training with shorter time series. Hence, for the training of the transfer learning LSTM architecture, 60% of the dataset (Second inflow case, presented in Figure 7) is fed to

the network, where 40% is left for testing to ensure a more definitive assessment of the training.



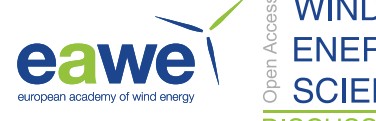

**Figure 9.** Distribution of the output tensors of each hidden LSTM layers for the First Model_1 and the Second Model_2 LSTM models, visualized via TensorBoard. Each slice displays a single histogram updated at each iteration. The 'oldest' iterations are further back and darker, while the 'newer' ones are lighter and closer to the front. The y-axis indicates the relative time of each update.

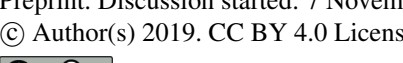



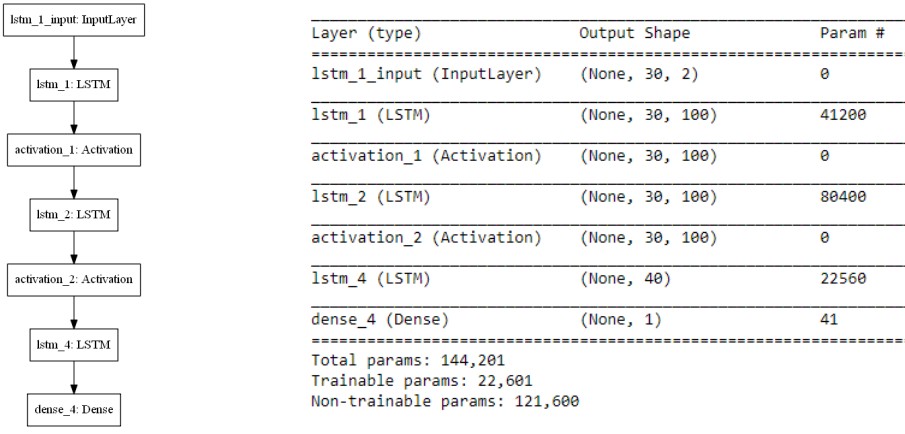

**Figure 10.** Architecture of the Transferred LSTM model for lag = 29s. *Hidden Layers:* `lstm_1`, `lstm_2`, `lstm_4`. *Output Layer:* `dense_6`. *Frozen Layers:* `lstm_1`, `lstm_2` (same layers as in the First LSTM model in Figure 5). *Trainable Layers:* `lstm_4`, `dense_4`. tanh is used as the activation function in both of the activation layers.

Apart from update of the weights in the last LSTM and the output layers (lstm_4 and dense_4 in Figure 10, respectively), none of the other hyper-parameters were changed in the training process of the Transferred LSTM model. This provides a certain repeatability to the training process, where the last two layers can be updated when a new flow case is encountered by the turbine. It is particularly an important feature for the control implementation as it enables the online learning and continuous

improvement of the model.

To put the performance of the Transferred LSTM model to test, the same validation case as in the Second LSTM model in Figure 8 is considered. This time, the estimations from the operation dependent Wind Observer (WSO) approach (described in Section 2) are also compared with the Transferred LSTM model. The time series in Figure 11(a) illustrates the sensitivity of the WSO estimations to the operation strategy under 40% down-regulation; where Op#1 curtails the turbine with constant rotational

speed $\Omega$, Op#2 uses the strategy with maximum $\Omega$ and Op#3 conducts a down-regulation strategy following the maximum thrust coefficient, $C_t$, curve. In Figure 12(a), (b) and (c), the error distributions of the WSO estimations under those three operational strategies are presented. While the overall performance of all WSO estimations are highly compelling, the results also indicate up to 4% variation in the mean bias of the WSO model. With the minimum mean error of 0.26%, WSO estimations with Op#2 are also compared with the Transferred LSTM model in Figure 11(c). For the investigated setup with DTU 10MW reference

turbine model fully recognized, the WSO results generally suggest a better agreement with the *true* available power, quantified in Figure 12(d). However, the Transferred LSTM model outperforms the Second LSTM model where more than 3% model bias is eliminated compared to Figure 8(d). This improvement is very promising for the implementation of the transfer learning for modelling high frequency time series with LSTM networks. Furthermore, the results also show the potential of such a deep learning approach for eliminating the operational dependencies of dynamic delta control with relatively low uncertainties. The

adaptation capabilities of transfer learning is to be tested with additional flow cases in the next sections.

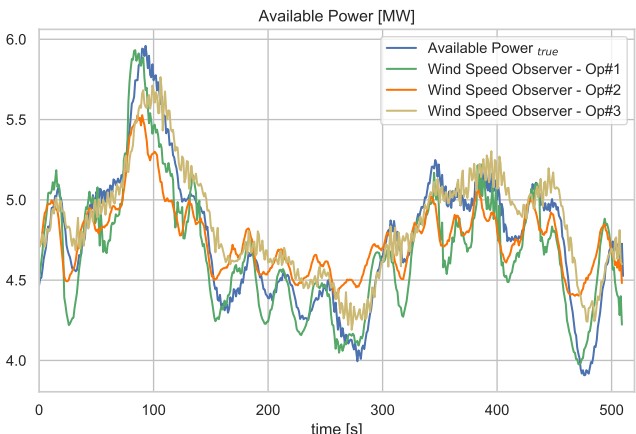

**(a)** Wind Speed Observer Available Power Estimation for 40% down-regulation case under 3 different curtailment strategies Op#1, Op#2, Op#3. Op#1 with constant rotational speed, Op#2 with maximum rotational speed and Op#3 follows minimum $C_t$ of DTU 10MW turbine.

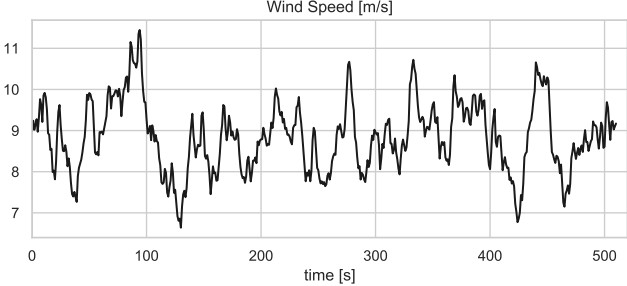

**(b)** 1-sec wind speed time series for mean wind speed = 9 m/s and mean TI = 10% validation case.

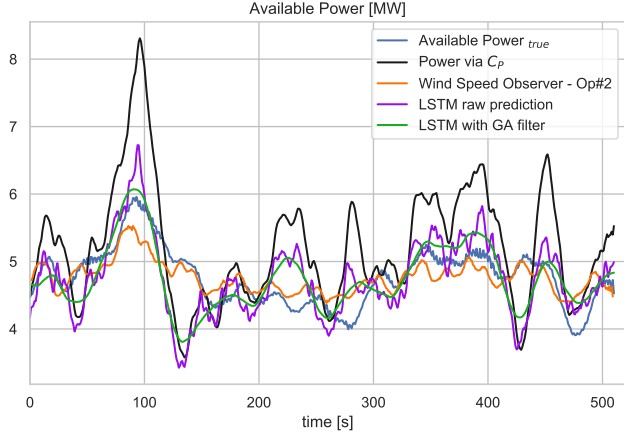

**(c)** LSTM model results with transfer learning Available Power Estimation (no dependency on level of curtailment). The raw results are presented as well as the post-processing with Gaussian convolution filter.

**Figure 11.** 1-Hz Time Series Comparison of Available Power of DTU 10MW turbine, estimated by (a) the Wind Speed Observer, see Section 2, using 3 different curtailment strategies Op#1, Op#2 and Op#3. (b) The comparison of the available power estimated via Wind Speed Observer Op#1, LSTM model with transfer learning from lower wind speed to higher wind speed case and direct $C_P$ approach.





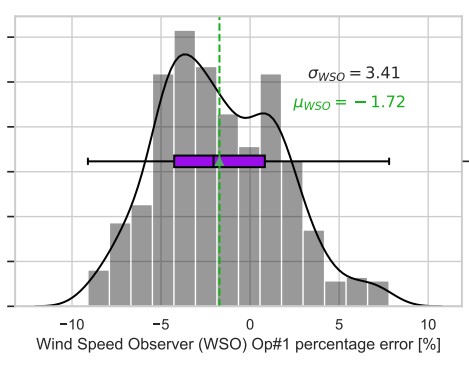

**(a)** Wind Speed Observer Op#1

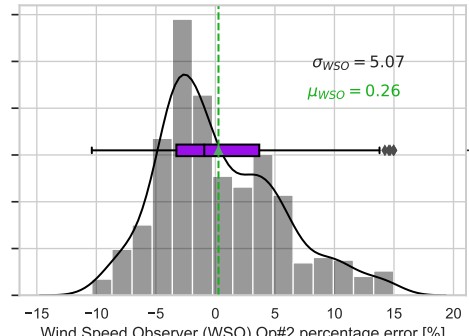

**(b)** Wind Speed Observer Op#2

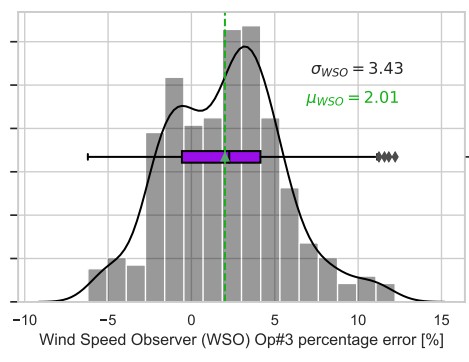

**(c)** Wind Speed Observer Op#3

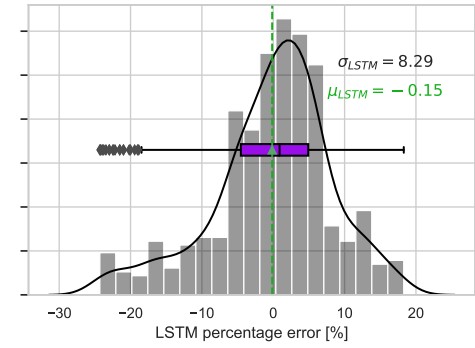

**(d)** LSTM model with transfer learning and Gaussian Convolution

**Figure 12.** 1-Hz percentage error distribution of the Available Power Estimation of 10MW turbine for mean wind speed = 9 m/s and mean TI = 10%, flow case represented in Figure 7. The presented performances belong to Wind Speed Observer approach in Section 2 different operational strategies. (a) Op#1: Constant rotational speed $\Omega$, (b) Op#2: Maximum rotational speed $\Omega$, (c) Op#3: Minimum thrust coefficient $C_t$ as 40% curtailment strategy; (d) LSTM model with transfer learning from lower wind speed to higher wind speed case (no dependency on the curtailment strategy).

### 3.1.4 Further Transfer Learning to Higher Wind Speed Flows

With the comparable results of the model-free transfer learning LSTM networks to the model-dependent WSO approach, even with potentially lower uncertainties in the simulation environment compared to the field implementation, here we test the approach for even higher wind speed flows. The First LSTM predictions were built and tested on 7m/s mean wind speed, where its information from the first two layers are then transferred to estimate the available power for 9m/s mean wind speed case. Here we further update the LSTM network to extend the training (and validity) domain to 11 m/s mean wind speed range, using the generated time series in Figure 13. Note that for all three steps of the learning, the mean TI remains 10% to isolate the effect wind speed on the network performance.




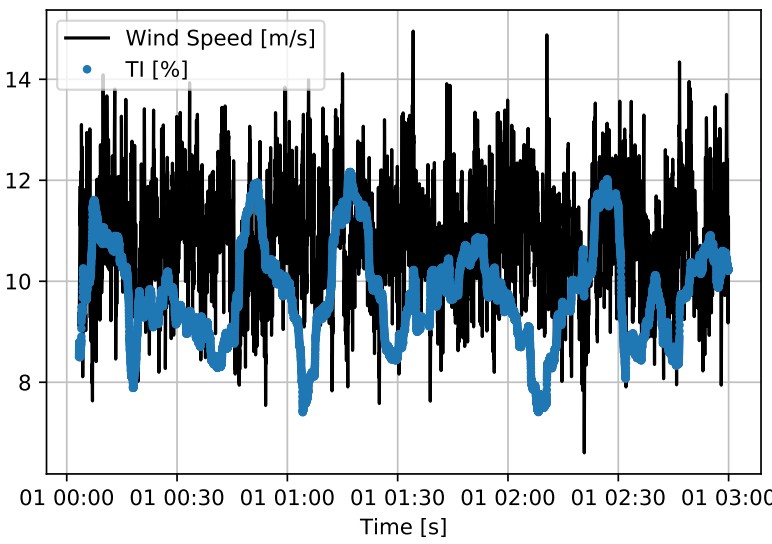

**Figure 13.** LSTM model training with further transfer learning input time series generated by HAWC2, down-sampled to 1-Hz. Mean Wind Speed = 11 m/s, Mean TI = 10%.

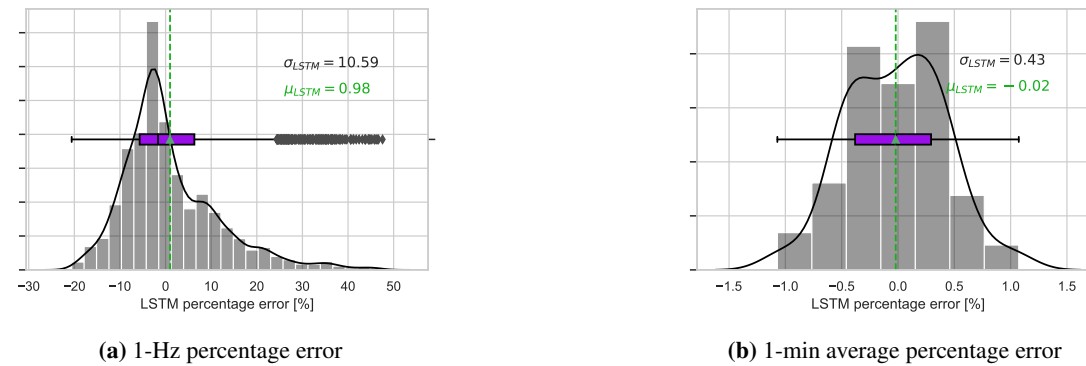

**(a)** 1-Hz percentage error          **(b)** 1-min average percentage error

**Figure 14.** LSTM model with Gaussian convolution filter prediction error for further transfer learning flow case. Mean Wind Speed = 11m/s and TI = 10% over 60-min validation case. (a) 1-Hz prediction error, (b) 1min average prediction error.

The resulting network with further transfer learning for 11 m/s mean wind speed (in Figure 14(a)) performs similar as in 9 m/s (in Figure 12(d)) and 7m/s (in Figure 6(d)) mean wind speed cases with less than 1% second-wise percentage error on average. Distinctly from the previous inflow cases, the tail towards the positive percentage error is longer in the final 1-Hz error distribution in Figure 14(a). This is mainly due to the fact that DTU 10MW reference turbine (with rated wind speed 11.4 m/s) occasionally enters the rated region according to the turbulent fluctuations around 11 m/s mean wind speed. Nevertheless, the variability of the model prediction error is significantly reduced when averaged for 1-min scale in Figure 14(b). The results




show that the model-free tranfser learning approach easily comply with the strictest TSO requirements in provision of the available power signal; *i.e.* the standard deviation of the 1-min percentage error of the available power is required to be less than ± 3.3% as stated in (50Hertz, Amprion, Tennet, TransnetBW, 2016).

### 3.1.5   Model performance on Higher Turbulence Intensity

As stated earlier, for all three inflow cases where the First LSTM network is generated and extended via transfer learning, the mean turbulence intensity remained TI = 10%. Here in this section, the models are tested under higher turbulence intensity (TI = 20%) with the same corresponding mean wind speed. Note that the generated network structures, *i.e.* the First LSTM model (7 m/s mean inflow speed), transfer learning LSTM model (9 m/s mean inflow speed) and further transfer learning LSTM model (11 m/s mean inflow speed), are not updated for higher TI cases. In other words, here we aim to test the capability of the trained networks under higher TI with the same mean wind speed for the inflow, without further model update.

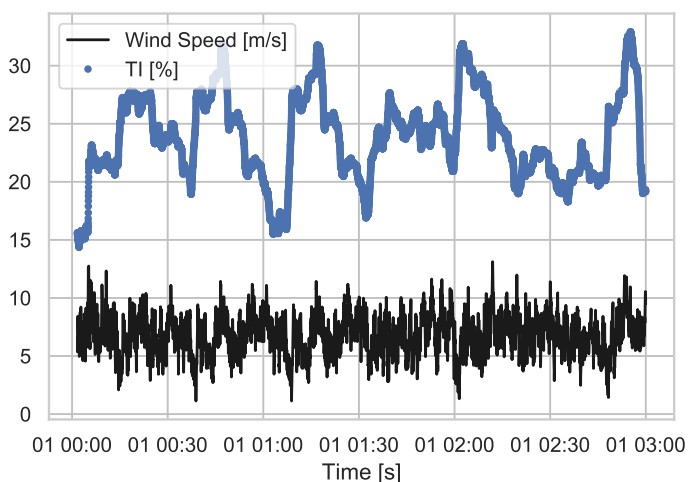

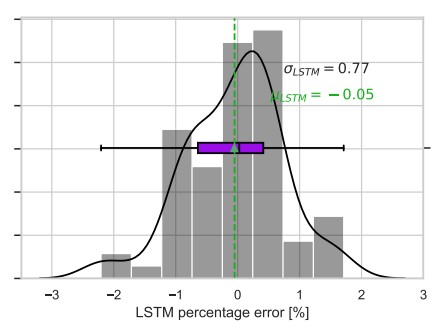

**(b)** 1-min average percentage error of the First LSTM model under higher TI

**(a)** Time series of the inflow dataset for higher TI validation of First LSTM model, trained on Mean Wind Speed = 7 m/s and TI = 20%.

**Figure 15.** Input time series and 1-min prediction error of the First LSTM model with Gaussian convolution filter, higher turbulence intensity flow case. Mean Wind Speed = 7 m/s and TI = 20% over 60-min validation case.




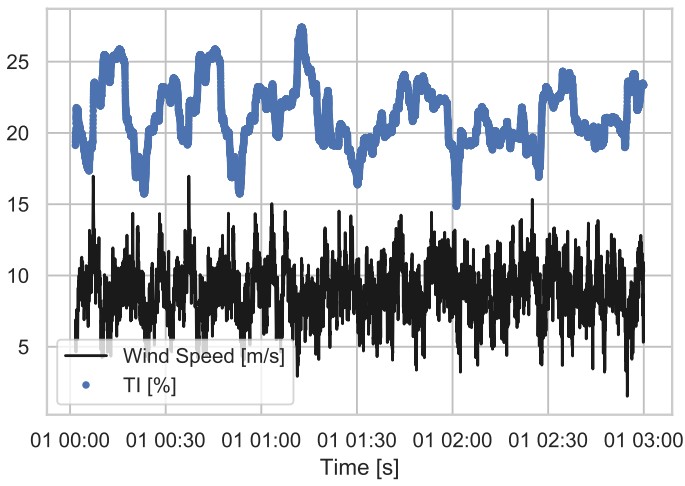

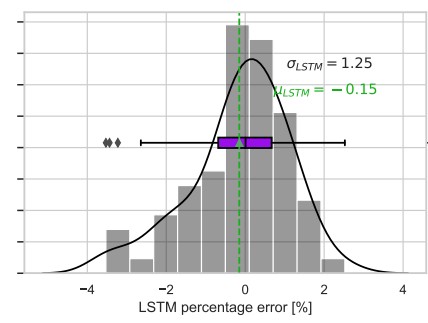

**(b)** 1-min average percentage error of the transfer learning LSTM model under higher TI.

**(a)** Time series of the inflow dataset for higher TI validation of the transfer learning LSTM model, updated for Mean Wind Speed = 9 m/s and TI = 20%.

**Figure 16.** Input time series and 1-min prediction error of the transfer learning LSTM model with Gaussian convolution filter, higher turbulence intensity flow case. Mean Wind Speed = 9 m/s and TI = 20% over 60-min validation case.

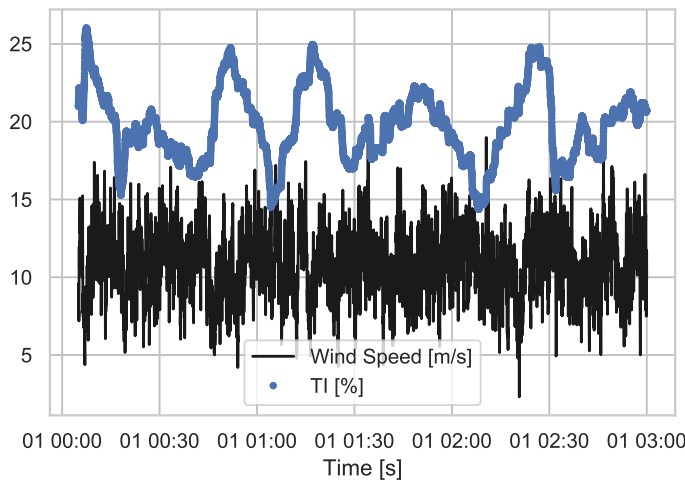

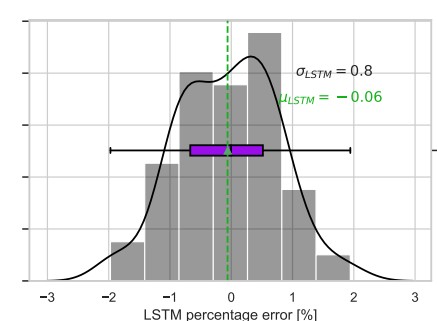

**(b)** 1-min average percentage error of the further transfer learning LSTM model under higher TI.

**(a)** Time series of the inflow dataset for higher TI validation of the further transfer learning LSTM model, updated for Mean Wind Speed = 11 m/s and TI = 20%.

**Figure 17.** Input time series and 1-min prediction error of the transfer learning LSTM model with Gaussian convolution filter, higher turbulence intensity flow case. Mean Wind Speed = 11 m/s and TI = 20% over 60-min validation case.





Figures 15, 16 and 17 focus on highly turbulent inflow cases (TI = 20%) and show the corresponding performance of the
three neural networks trained and updated for increasing wind speeds via transfer learning. It is seen that the 1-min average
prediction errors of the models are consistently low for highly turbulent flows as well; hence further training (or model update)
is not required. The maximum standard deviation of 1-min averaged percentage error is 1.25% with the largest bias of 0.15%.
Accordingly, it can be said that the sensitivity of the networks to changing wind speed is much higher than the turbulence, and
the updates are to be implemented solely based on the altering inflow velocity which is likely to reflect a different operational
region. The available power prediction of the described LSTM architecture and the updating scheme with 2 m/s wind speed
increase (7 m/s – 9 m/s – 11 m/s) is shown to easily comply with the strictest grid code requirements under different turbulence
realizations.

## 4   Conclusions

The dynamic estimation of available power of a wind turbine is essential both for power system stability and marketability
of the reserve power. The current estimations are highly sensitive to the down-regulation strategy and prone to turbine model
uncertainties and inadequacies. Here we propose a purely data-driven, model-free methodology based on Long Short Term
Memory (LSTM) neural networks. This state-of-the-art deep learning architecture is implemented to map the available power
of DTU 10MW reference turbine under turbulent inflow generated in HAWC2. The trained networks are adapted to the changes
in incoming mean wind speed via transfer learning, where the parameters only in the last layer are updated when the new inflow
information is available.
    The First LSTM network has 3 hidden layers with 100, 100 and 40 neurons respectively, which is trained using 1-Hz power
output under normal operation with 7 m/s mean wind speed and 10% turbulence intensity (TI). Validation on a separate 10-min
flow case with the same mean wind speed and TI shows less than 0.5% bias and the distribution of 1-sec percentage error is
further narrowed with running a Gaussian convolution filter over it. The resulting error distribution of second-wise estimated
available power has less than 8% standard deviation with the same bias. Same architecture is used to train the Second LSTM
network with increase in mean wind speed to 9 m/s and same TI level of 10%. Although the width of the distribution is similar,
the bias has increased to more than 3%, indicating the need to re-tune the hyperparameters of the architecture. In fact, the
comparison of the fitted parameters between the First LSTM and the Second LSTM networks for each layer shows analogous
distributions of the weights. This further motivates the transferability of the learnings of the first two LSTM layers, where only
the parameters of the last layer need to be updated for the changing incoming mean wind speed. With a significant reduction
in the number of parameters to fit, the Transferred LSTM network has the capability of faster and more robust training, even
with limited data. The performance of the Transferred LSTM network is also evaluated using a separate 10-min time series
with 9 m/s mean wind speed and 10% TI. The results are very comparable with the outcome of the First LSTM model,
which demonstrates the adaptability of the network to changing inflow conditions with the update of the last LSTM layer. The
Transferred LSTM also outperforms the Second LSTM network with a significant decrease in bias ( -0.15%), eliminating the
need to re-tune the hyperparameters or developing a new network structure from scratch.





The Transferred LSTM network is also compared with the model and operation dependent Wind Speed Observer (WSO) approach. For the investigated setup where the DTU 10MW reference turbine model fully transparent, the WSO results generally

suggest a better agreement with narrower 1-Hz percentage error distributions. However, the sensitivity of the WSO approach to the curtailment strategy is also clearly seen as the results indicate up to 4% variation in the mean bias of the WSO model. The uncertainty of the approach is expected to grow further under the field conditions where there is a potential lack of detailed information regarding the operation strategy and manufacturer-calibrated power coefficients which are generally unable to account for variability influenced by local conditions.

To ensure the applicability of the transfer learning to several inflow cases, the approach is tested for even higher wind speed flows. Further Transferred LSTM network is trained only to update the last LSTM layer with 11 m/s mean wind speed and 10% TI case, where the first two layers are coming from the First LSTM model with 7 m/s mean wind speed validity domain for the same TI. The performance of the Further Transferred network is evaluated within the framework of strict grid requirements, where the quality of the available power signal is to be assessed at 1-min intervals with required accuracy of less than 3.3%

standard deviation of the error distribution. Corresponding 1-min average percentage error of the Further Transferred network indicates easy compliance with the regulations, with both bias and standard deviation less than 0.5%. Similar agreement is observed when all the networks (*i.e.* First LSTM with 7 m/s wind speed, Transferred LSTM with 9 m/s wind speed and Further Transferred LSTM with 11 m/s wind speed) are tested under higher TI of 20%, indicating the robustness of the developed algorithm.

Finally, it should be noted that the neural networks with transfer learning ability used in this study can easily be implemented in operating wind turbines in the field. The second-wise wind speed input to the approach can be provided either from the standard nacelle anemometers or additional sensors such as meteorological masts or remote sensors (*e.g.* lidars, radars, etc.). This study is a conceptual evidence that well-trained neural networks can be applied to determine the set-point for implementing delta control, or to assess the level of reserves when using balance control, even with limited information in the field conditions.

The transferability of the network adds the ability for online learning which ensures the continuous improvement of the model-free available power estimation.

The networks developed in this study can be extended to forecast applications, where the input that is read throughout the hindsight horizon (*e.g.* 29-seconds for the cases presented here) is used to predict the available power in forecast horizon longer than 1-second (*e.g.* 1-minute ahead). Similarly, the approach can be implemented to several turbines within the wind

farm. For this configuration, the wind direction should be defined as an additional input to take the correlations of local wake effects and power into account. Finally, the neural network algorithm can be updated given the developments within the deep learning research over time. The advancements in the sequential processing (*e.g.* convolutional LSTMs where the internal matrix multiplications are exchanged with convolution operations, gated recurrent units (GRUs) where the 3-gated LSTMs are 'simplified' with an update and a reset gate, etc.) can easily be utilised when beneficial, keeping the approach up-to-date.



*Code and data availability.* Both the data and the script to train the networks (First LSTM and Transfer Learning LSTM networks) can be reached via the project at https://gitlab.windenergy.dtu.dk/tuhf/deep-learning-for-available-power-estimation

## Appendix A:  Training history of the LSTM networks

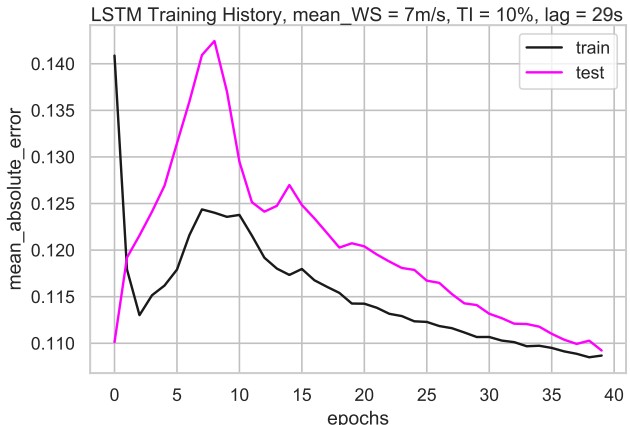

**Figure A1.** Training history of the First LSTM network with mean wind speed = 7 m/s, mean TI = 10%, time lag = 29s. Early stop at 37 epochs due to convergence. The network architecture is presented in Figure 5, batch size = 60

*Author contributions.* The LSTM network training as well as the transfer learning implementation is performed by Tuhfe Göçmen. The need for such an approach is underlined by Albert Meseguer Urban who also generated the time series in DTU 10MW turbine model in
HAWC2. Jaime Liew also contributed in defining the research gaps in the field and developed the implementation cases for LSTM network as the set-point for curtailment in HAWC2 controller, therefore also ensured the generalizability and validity of the code. Alan Wai Hou Lio contributed with Wind Speed Observer and the sensitivies of the current methods for available power estimation at the single turbine level.

*Competing interests.* The authors declare that they have no competing interest.

*Acknowledgements.* The study is partially supported by the CONCERT Project (Project no. 2016-1-12396), funded by Energinet.dk under
the Public Service Obligation (PSO), by TotalControl (Project no. 727-680) within the Horizon2020 framework funded by the European Union and the PowerKey Project (Project no. 64017-0045) under the EUDP Program.



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
