# Peer review of "Model-free Estimation of Available Power using Deep Learning"

_Wind Energy Science, 2019_

## Short Comment (SC1) · 9 Mar 2020

The paper entitled "Model-free Estimation of Available Power using Deep Learnin" presents an interesting topic to the journal "Wind energy Science". The paper cannot be accepted in the present form, please find as follow my comments: Major • The paper presents some plagiarism issue, not for rejecting it, but I suggest to correct it because it looks like copy and paste with minor changes: Lines 16-26; 45-60; 74-78; 84-96; 178-179; 191-195 • The abstract shows what the paper describes, it is good, but it does not show the novelty of the paper. • The main novelties should be also shown clearly in the introduction • I have not seen the keywords in the paper. • To describe all the variables mentioned in the equations employed • To describe and justify the hiper-parameters of the NN employed ("the final network has 3 hidden

layers with 100, 100 and 40 LSTM neurons, as indicated in Figure 5.") It is not clear enough. • To include Appendix A to the text of the paper. The curves shapes are a bit strange from my experience, are they correct? I do not say that they are wrong. • Please detail clearly how the accuracy is calculated, and where is obtained the accuracy of 1% mentioned in the abstract Minor • Do not use acronyms in the abstract • Use the acronyms origin before to use them, of to use the acronym when they are mentioned before, and do not repeat, e.g. turbulence intensity. To revise all the paper carefully. • What does DTU mean? Technical University of Denmark? • Terms as "heavily" are not correct for a scientific paper. • Do not include a large number of references together. They should be reduced or describe their contribution in the paper individually. See e.g. "e.g. Ministere des Affaires Economiques (2019); Energinet.dk (2017); 50Hertz, Amprion, Tennet, TransnetBW (2016); 20 Elia, Belgium (2015); EirGrid (2015); National Grid (2014)).": 1-2 is good, could be until 3 in certain cases, no more. • "or as delta control" not correct • "The wind turbine operator can then announce its participation in the reserve market online or ahead of time with the intention of performing delta or balance control." To rewrite it, it is confuse • What is U in equation 1?

Finally, just curious, how have you plotted Fig. 6b-d?

---

## Referee Comment (RC2) · Martin Felder (Referee) · 25 Jun 2020

This paper adresses the problem of assessing the instantaneous power available at a wind turbine from the inflow average wind speed and turbulence intensity. The goal is to provide accurate 1 min estimates of available power, to compare against actual power delivered in cases of curtailment. Classical, model-based algorithms are compared against a novel model free method based on LSTM type neural networks. The whole study is based purely on simulated data, downsampled to 1 Hz resolution.

Overall, while the method described does seem to solve the problem, some basic rules of neural network training were followed only haphazardly, if at all. The motivations for some of the decsions made in the process (e.g. early stopping, network size, postprocessing) are not clear. I am aware that this is not a machine learning journal, but throwing a bunch of data into a ML toolbox and applying some new buzzword method to them is not good science. Before moving on to advanced methods like transfer learning, the authors would do well to better understand the basic model first, and find out what the limitations of a well trained single neural network really are. Because from a practical standpoint, training a model once and applying it in an unchanged fashion operationally is much easier than implementing a fault tolerant re-training strategy of any kind. At least it would be helpful to know how a properly tuned standard machine learning method, like a single network trained on data from all wind regimes, performes against the its transfer-learned competitors. So, in my opinion this paper needs some major rework in order to be considered for publication.

Detailed comments follow:

The citations in the Introduction should be balanced an little more. Despite the paper's focus on ML, there are only two relatively old (considering the dynamics of the field) references to ML use in wind power forecasting (lines 40 and 42), while we find about 15 citations related to grid codes and curtailment, and ∼10 on power curve modelling.

On a similar note, it seems questionable to discuss EKF application to the turbine model on three pages, including mathematics, while only showing one sketch of an LSTM neuron, with no explanation on how these models are actually trained. There is no mention of the loss function or the training algorithm used. At this point Appendix A definitely needs to be included and discussed. For example, why is the perfomance on the test dataset at the start of training about the same as after training? Why is the training stopped at epoch 37, while the test error is obviously still decreasing after a small bump? Why do you need LSTM at all, since your best lag comes out as 29 - hardly requiring a "long" memory. Also, in Figure A1 please start the ordinate at zero, otherwise the plot scale is misleading.

Figure 5 should be reduced to a well formatted table: The TensorFlow model dump

contains misleading information (e.g. what means "None" for the time step dimension? Why does "activation" have a shape but no parameters? ...). The block diagram is trivial and therefore redundant. [FS] das ist ein Screenshot von Keras model.summary() und der Plot der raus kommt wenn man in Keras plot_model() macht. Auf der einen Seite kann das jeder lesen und weiß was das None Zeug bedeuten soll, auf der anderen Seite ist es etwas lazy ;) Da es die Keras Leute aber alle so kennen kann man das imho schon so durchgehen lassen.

In line 226, "test dataset" is used to refer to the part of the data which is utilized for determining the early stopping point during gradient descent. Since several years, it has been common practice in the ML literature to call this the "validation dataset". The "test dataset" would then be the shorter time series mentioned in line 227. While I'm personally not happy with this nomenclature, I strongly suggest adhering to the quasi-standard here, to avoid further confusion.

The definition of "lag" needs to moved from line 248ff to the start of Sec. 3.1, otherwise the description of preprocessing is hard to understand.

Line 262ff: It seems very suspicious to me that the simple application of a Gaussian smoother improves your prediction. What information are you adding here that the LSTM model does not have? And why does the LSTM not have it? Is there so much random noise in the network output? Is the loss function for the training exactly the same as the one you use for estimating the test set errors? What is the width of the Gaussian smoother, and how was it determined? By introducing a smoothing function, you introduce correlations between the errors of many neighboring 1 Hz samples. Hence it could be argued that the statistics presented are no longer comparable, i.e. you should smoothe the results from the model based methods by the same Gaussian. Furthermore, the effective number of samples you compare against is reduced by a factor corresponding to the width of the filter. This raises the question of whether or not the interval used for evaluation is long enough to draw some of the conclusions in the paper.

**WESD**

Interactive comment

Line 280ff: The only hyperparameter "optimized" (if you can call a 5 point 1-dim grid search "optimizing") for the low wind speed case is the lag, since the rest of the architecture was apparently derived from some rule of thumb, with no further explanation given. It is good practice to at least check a few combinations of network depth and width, including extremely small networks. These may obviate the need for transfer learning and/or output smoothing altogether.

Line 284: Why is it not possible to perform automatic hyperparameter search when the training time for the full model is only in the order of an hour? This kind of optimization is standard procedure in many ML applications.

In Figure 11, a) and b) are described, but not c). Designating the operation modes introduced in Fig. 2 as Max-Omega, Const-Omega and Min-Cp now as Op#1 to 3 is confusing. Please stick to one nomenclature. The comparison between a) and c) would greatly benefit from not repeating the "Power via Cp" curve from Fig. 8, but instead using the same vertical scale in both graphs.

Fig. 13/14: Now you are apparently using a 60 min timeseries for evaluation, while before it was 10 min? Since Fig. 13 and others of the same kind have unclear abscissa labels (not [s], but days and hours?), it is not clear anymore which data are used for what. Please clarify.

Since the focus of the investigation is the adherence to the 1 min/3.3% error grid code, and the errors against the 1 min timeseries are much smaller than the ones against the 1 Hz timeseries, it would be helpful to see the performance of the model based algorithms against the 1 min resolution timeseries as well, because this is what we are eventually interested in.

Another point that needs to be discussed before talking about practical applications is the validation of the algorithm on real observations, as compared to a pure simulation. Training neural networks on essentially noise-free input data is of course much less problematic than dealing with real-world data issues. In fact, the authors claim that

Discussion paper

[Figure]

C4

their method is superior to model-based approaches because of not explicitly relying on manufacturer data, but there is no proof to that claim. If no real-world data are available for testing, the claim could be corroborated for instance by simulating changes to Cp and observing the result on neural network performance.

---

## Author Comment (AC1) · 29 Jul 2020

Dear reviewers Fausto Pedro García Márquez and Martin Felder,

Thank you very much for taking the time to review as well as your valuable comments and suggestions for the revision of our article. We have taken all of them into consideration to improve our work and prepared a revised version of the manuscript accordingly. Please find specific replies and pointers to your comments down below.

Many greetings,

Tuhfe Göçmen, Albert Meseguer Urbán, Jaime Liew, and Alan Wai Hou Lio
* * *
**Fausto Pedro García Márquez**

**Major:**

1. The paper presents some plagiarism issue, not for rejecting it, but I suggest to correct it because it looks like copy and paste with minor changes: Lines 16-26; 45-60; 74-78; 84-96; 178-179; 191-195

We believe what is referred here is not plagiarism (from other works) but a repetition of certain parts of the text within the manuscript. They are often included to emphasise a message further. Regardless, the indicated paragraphs are slightly rephrased to avoid 1-1 recurrence within the article.

2. The abstract shows what the paper describes, it is good, ´ but it does not show the novelty of the paper. The main novelties should be also ´ shown clearly in the introduction, I have not seen the keywords in the paper

The novelty of the manuscript originates from its multi-disciplinary character. The fact that it combines wind turbine control considerations and state-of-the-art forecasting tools to develop a model-free approach with online learning capabilities is novel. Additionally, it evaluates and validates the tool from power systems perspective with quantified grid code compliance, which is often overlooked. To reflect that, a few sentences have been added to the abstract, which are also included in the Introduction section.

The keywords section could not be located in the manuscript template, but could be added editorially if needed.

3. To describe all the variables mentioned in the equations employed

They are all now described, including the newly added set of equations for LSTM formulation.

4. To describe ´ and justify the hiper-parameters of the NN employed ("the final network has 3 hidden layers with 100, 100 and 40 LSTM neurons, as indicated in Figure 5.") It is not clear enough.

A selective grid search is presented in Table 1 in the revised manuscript and corresponding discussions are now included in the paragraph before/after Table 1.

5. To include Appendix A to the text of the paper. The curves shapes are ´a bit strange from my experience, are they correct? I do not say that they are wrong.

It is now Figure 5 in the text. The behaviour of the training history is now plotted for further epochs, which we hope makes trends a bit clearer. As included in the response to Reviewer #2 Martin Felder, the low validation loss at the very first epoch is still interesting – further details for that can be read further down in the corresponding response #4 to Reviewer #2.

6. Please detail clearly how the accuracy is calculated, and where is obtained the ´ accuracy of 1% mentioned in the abstract

For the First, Second, Transfer and Further Transfer LSTM networks the mean of the 1-sec and 1-min error distribution is less than 1% in almost all the test cases, except for 1 (with bias = 1.28%) - see Figures 6c, 11d, 14b, 16b and 17b for the presented test cases with varying inflow wind speed and TI levels.

**Minor:**

7. Do not use acronyms in the abstract. Use the acronyms origin before to use them, of to use the acronym when ´ they are mentioned before, and do not repeat, e.g. turbulence intensity. To revise all the paper carefully

When it comes to LSTM, it simply is a convention to use the abbreviation rather than 'long-short-term memory' within literature. Accordingly, LSTM is simply better known (and more searched in scientific literature databases) than its longer name and always referred accordingly in the other abstracts in relevant fields.

8. What does DTU mean? Technical University of Denmark?

Yes 😊 it is added to author affiliations as well. DTU 10MW follows the same convention with other reference wind turbines such as NREL 5MW.

9. Terms as "heavily" are not correct for a scientific paper

Substituted with highly.

10. Do not include a large ´ number of references together. They should be reduced or describe their contribution in the paper individually. See e.g. "e.g. Ministere des Affaires Economiques (2019); Energinet.dk (2017); 50Hertz, Amprion, Tennet, TransnetBW (2016); 20 Elia, Belgium (2015); EirGrid (2015); National Grid (2014)).": 1-2 is good, could be until 3 in certain cases, no more

The mentioned reference on page 1 is omitted.

11. "or as delta control" not correct

'either balance control, …, or delta control, …' now changed to: 'both balance control, …, and delta control, …'

12. "The wind turbine operator ´ can then announce its participation in the reserve market online or ahead of time with the intention of performing delta or balance control." To rewrite it, it is confuse

Elaborated slightly as: '… Accordingly, the wind turbine operator can announce its participation in the reserve market online or ahead of time with the intention of performing delta or balance control for curtailment'

13. What is U in equation 1?

(effective) wind speed, now added below equation 1.

14. Finally, just curious, how have you plotted Fig. 6b-d?

The figures are generated by combining distplot and boxplot functions from Seaborn library in Python – see https://seaborn.pydata.org/generated/seaborn.distplot.html, and https://seaborn.pydata.org/generated/seaborn.boxplot.html.

**Martin Felder**

1. The citations in the Introduction should be balanced an little more. Despite the paper's focus on ML, there are only two relatively old (considering the dynamics of the field) references to ML use in wind power forecasting (lines 40 and 42), while we find about 15 citations related to grid codes and curtailment, and ~10 on power curve modelling.

Agreed – now the citations are updated to reflect more recent studies, also added articles that are published after the original submission of the manuscript.

2. On a similar note, it seems questionable to discuss EKF application to the turbine model on three pages, including mathematics, while only showing one sketch of an LSTM neuron, with no explanation on how these models are actually trained.

The section on LSTM description is elaborated in the revised version. Mathematical procedure to update the cell state and define the final output of an LSTM neuron is presented.

3. There is no mention of the loss function or the training algorithm used

Adam optimisation algorithm and mean absolute error loss function are implemented for the training process - Information is now added to newly Figure 5 (previously in Appendix) training history.

4. At this point Appendix A definitely needs to be included and discussed. For example, why is the perfomance on the test dataset at the start of training about the same as after training? Why is the training stopped at epoch 37, while the test error is obviously still decreasing after a small bump?

Appendix A is now Figure 5 – the early stop is not implemented any longer and the model training is shown until epoch = 120. Accordingly, it is clarified now that the validation loss is lower in the later epochs than the very first epoch.  This low validation loss at the very first epoch is still interesting and found to be consistent for most of the architectures trained in Table

1. On the other hand, the trend disappears for other loss functions than MAE, *e.g.* MAPE which is not used in this study since it is based on the scaled parameters and the resulting normalised losses do not translate/agree well with the rest of the percentage error evaluations in the paper. Therefore, since it clearly does not indicate an overall higher accuracy, the network is trained further to its fuller potential using MAE as the loss function.

5. Why do you need LSTM at all, since your best lag comes out as 29 - hardly requiring a "long" memory. Also, in Figure A1 please start the ordinate at zero, otherwise the plot scale is misleading.

Both Table 1 (LSTM vs FFNN) and Table 3 (4s lag vs. 29s lag) in the revised version point to added benefits of using neurons with memory. Although 29s might not seem as much, it is still 29-steps lag for our 1Hz predictive network and covers about half the evaluation period as set by the grid codes.

Regarding the comment about the scale of the plot - since the difference between training and validation losses are small, it becomes hard to see if the training history plot y-axis ordinates at zero. For your convenience, we have put the example here. However, we keep the 'zoomed-in' version in the paper for enhanced clarity.

[Figure]

6. Figure 5 should be reduced to a well formatted table: The TensorFlow model dump contains misleading information (e.g. what means "None" for the time step dimension? Why does "activation" have a shape but no parameters? ...). The block diagram is trivial and therefore redundant. [FS] das ist ein Screenshot von Keras model.summary() und der Plot der raus kommt wenn man in Keras plot_model() macht. Auf der einen Seite kann das jeder lesen und weiß was das None Zeug bedeuten soll, auf der anderen Seite ist es etwas lazy ;) Da es die Keras Leute aber alle so kennen kann man das imho schon so durchgehen lassen.

It is now formatted as Table 2. Similarly for transfer learning LSTM architecture on Table 5.

7. In line 226, "test dataset" is used to refer to the part of the data which is utilized for determining the early stopping point during gradient descent. Since several years, it has been common practice in the ML literature to call this the "validation dataset". The "test dataset" would then be the shorter time series mentioned in line 227. While I'm personally

not happy with this nomenclature, I strongly suggest adhering to the quasistandard here, to avoid further confusion.

Agreed – the nomenclature now follows the 'quasi-standard'.

8.  The definition of "lag" needs to moved from line 248ff to the start of Sec. 3.1, otherwise the description of preprocessing is hard to understand.

It is now moved to the second paragraph of Section 3.1

9.  Line 262ff: It seems very suspicious to me that the simple application of a Gaussian smoother improves your prediction. What information are you adding here that the LSTM model does not have? And why does the LSTM not have it? Is there so much random noise in the network output? Is the loss function for the training exactly the same as the one you use for estimating the test set errors? What is the width of the Gaussian smoother, and how was it determined?

By introducing a smoothing function, you introduce correlations between the errors of many neighboring 1 Hz samples. Hence it could be argued that the statistics presented are no longer comparable, i.e. you should smoothe the results from the model based methods by the same Gaussian. Furthermore, the effective number of samples you compare against is reduced by a factor corresponding to the width of the filter. This raises the question of whether or not the interval used for evaluation is long enough to draw some of the conclusions in the paper.

These are valid and fair points – thanks again for detailed input here. Especially the reduction of the effective number of samples used for evaluation is the attribute that would be the most concerning for the structure and the concluding remarks of the paper. As listed in the next response, a broader grid search (which is now presented in Table 1) as well as further training of the model for large number of epochs have indeed provided a better performing network that the output smoothing with Gaussian convolution is no longer needed. Corresponding discussions have been omitted from the manuscript.

10. Line 280ff: The only hyperparameter "optimized" (if you can call a 5 point 1-dim grid search "optimizing") for the low wind speed case is the lag, since the rest of the architecture was apparently derived from some rule of thumb, with no further explanation given. It is good practice to at least check a few combinations of network depth and width, including extremely small networks. These may obviate the need for transfer learning and/or output smoothing altogether.

As you mentioned earlier, the journal not being an ML based journal we have avoided to include the grid search performed. It is now selectively included in Table 1 for both LSTM and feed-forward networks for a few combinations. The neurons have 50 step increase per layer and accordingly the final selection of neurons for the last layer have been increased from 40 to 50 neurons. This update in 3-layer LSTM with 100-100-50 neurons have resulted in better validation loss, and overall better performance for the test dataset of the First LSTM network. The results indeed alleviate the need of applying Gaussian smoothing function, but transfer learning is still found to be beneficial.

11. Line 284: Why is it not possible to perform automatic hyperparameter search when the training time for the full model is only in the order of an hour? This kind of optimization is standard procedure in many ML applications.

As indicated, the full training of the final LSTM network with the optimal performance is in the order of an hour, where the smaller networks take a bit less time. Accordingly, performing an automatic hyperparameter search for number of layers, number of neurons per layer and ideal lag would take up to a day for a single flow case with a given wind speed when 20+ models are trained and compared against each other, depending on the combination of the gridded hyperparameter domain. This is clearly far from desired flexibility in the final application, where online training and continuous improvement in network performance is envisaged during the operation of the turbine. We should also note that, since the transfer learning is updating the last layer of the weights using the latest normal operation data, potential degradation in turbine available power (both small and large scale, temporally) is also taken into account.

The phrase '…It indicates the need to specifically tune the hyper-parameters for each separate flow case. It is a cumbersome process with very little room for automation.' is now changed to 'It is a cumbersome process with high computational cost.' to more clearly state the issue.

12. In Figure 11, a) and b) are described, but not c). Designating the operation modes introduced in Fig. 2 as Max-Omega, Const-Omega and Min-Cp now as Op#1 to 3 is confusing. Please stick to one nomenclature. The comparison between a) and c) would greatly benefit from not repeating the "Power via Cp" curve from Fig. 8, but instead using the same vertical scale in both graphs.

In Figure 11, all the subfigures a), b) and c) are now described. The reference for the control strategies are now unified as suggested, *i.e.* following Const-Ω etc. The comparison between Figure 11 a) and c) is performed in Figure 12 in terms of error distributions, therefore, although it changes the scale the 'Power via Cp' approach is kept in Figure 11 c) as it is still a widely implemented approach to estimate available power.

13. Fig. 13/14: Now you are apparently using a 60 min timeseries for evaluation, while before it was 10 min? Since Fig. 13 and others of the same kind have unclear abscissa labels (not [s], but days and hours?), it is not clear anymore which data are used for what. Please clarify.

A general description of the evaluation periods and frequency of error distributions is now added at the end of Section 3.1 (just before Section 3.1.1). Since the training parameters reduce significantly for the transfer learning, the training/validation split of the 3-hour dataset is changed from 80-20 to 60-40 which provides more than an hour of data to validate the Transferred and Further Transferred LSTM networks. This is already mentioned in Section 3.1.3 – now lines 305-310. The ambiguous x-axes for time labels are now corrected for all the time series plots to represent the steps in seconds.

14. Since the focus of the investigation is the adherence to the 1 min/3.3% error grid code, and the errors against the 1 min timeseries are much smaller than the ones against the 1 Hz timeseries, it would be helpful to see the performance of the model based algorithms against the 1 min resolution timeseries as well, because this is what we are eventually interested in.

Although agreed, for sharper focus of the article on LSTM performance evaluation, the discussions on model-based approach is kept limited to Section 3.1.3, hence on 1-sec basis within 10min test period. In Section 3.1.3, a well as the Introduction, there exist several arguments on why the focus of the article is model-free network – an additional argument being now Figure 12, following the suggestion raised in the next point of the review.

15. Another point that needs to be discussed before talking about practical applications is the validation of the algorithm on real observations, as compared to a pure simulation. Training neural networks on essentially noise-free input data is of course much less problematic than dealing with real-world data issues. In fact, the authors claim that their method is superior to model-based approaches because of not explicitly relying on manufacturer data, but there is no proof to that claim. If no real-world data are available for testing, the claim could be corroborated for instance by simulating changes to Cp and observing the result on neural network performance.

Thank you for the suggestion, that motivated us to include now Figure 12 and associated discussions on lines 355-360; as well as updated Figure 10(c). We have introduced ±5% uniform uncertainty to Cp for curtailment, where full availability of the information is assumed for max Cp, *i.e.* under normal operation. The results indeed point to significant deviations in the lowest biased performance for the model-based approach – further contributing to the arguments on high Cp sensitivity. That would clearly have no affect on the neural network performance as the wind speed and therefore the available power output of the turbine (*i.e.* the input and the output features for the network) would remain the same.